# Recent Advances Regarding Precious Metal-Based Electrocatalysts for Acidic Water Splitting

**DOI:** 10.3390/nano12152618

**Published:** 2022-07-29

**Authors:** Yuanting Peng, Yucong Liao, Donghao Ye, Zihan Meng, Rui Wang, Shengqiu Zhao, Tian Tian, Haolin Tang

**Affiliations:** 1Wuhan Marine Electric Propulsion Research Institute, Nanhu Qixiao, Wuhan 430064, China; 13907127990@139.com (Y.P.); 15071289880@139.com (D.Y.); 2Foshan Xianhu Laboratory, Advanced Energy Science and Technology Guangdong Laboratory, Xianhu Hydrogen Valley, Foshan 528200, China; liaoyucong@whut.edu.cn (Y.L.); mengzihan@xhlab.cn (Z.M.); rwang@whut.edu.cn (R.W.); zhaoshengqiu@whut.edu.cn (S.Z.); 3State Key Laboratory of Advanced Technology for Materials Synthesis and Processing, Wuhan University of Technology, Wuhan 430070, China

**Keywords:** electrochemical water splitting, precious metal-based electrocatalysts, acidic condition, oxygen evolution reaction, hydrogen evolution reaction

## Abstract

Electrochemical water splitting has wide applicability in preparing high-density green energy. The Proton exchange membrane (PEM) water electrolysis system is a promising technique for the generation of hydrogen due to its high electrolytic efficiency, safety and reliability, compactness, and quick response to renewable energy sources. However, the instability of catalysts for electrochemical water splitting under operating conditions limits their practical applications. Until now, only precious metal-based materials have met the requirements for rigorous long-term stability and high catalytic activity under acid conditions. In this review, the recent progress made in this regard is presented and analyzed to clarify the role of precious metals in the promotion of the electrolytic decomposition of water. Reducing precious metal loading, enhancing catalytic activity, and improving catalytic lifetime are crucial directions for developing a new generation of PEM water electrolysis catalysts. A summary of the synthesis of high-performance catalysts based on precious metals and an analysis of the factors affecting catalytic performance were derived from a recent investigation. Finally, we present the remaining challenges and future perspectives as guidelines for practical use.

## 1. Introduction

As the economy has evolved rapidly, fossil fuels such as coal, oil, and natural gas have been extensively used [1]. This has worsened pollution and the greenhouse effect, making the development of green and renewable energy sources the most effective means of addressing energy and environmental issues [2,3]. Hydrogen energy is the most promising secondary source of pollution-free energy in the twenty-first century to replace fossil fuels. With a specific energy density of 142.35 kJ/kg, hydrogen energy delivers unrivaled advantages in terms of cost and efficiency [4]. Hydrogen is abundant in water, and its use does not contribute to environmental contamination. The electrochemical water splitting (EWS) method for producing hydrogen and oxygen is advantageous due to its simplicity, low environmental impact, high efficiency, and high hydrogen purity [5]. Consequently, it is considered a cost-effective and promising method for producing hydrogen with ultra-high purity. The EWS process can be powered by additional renewable energy from sources such as wind and solar [6]. Hydrogen generation by water electrolysis using a proton exchange membrane (PEM) is the most promising EWS method [7]. In contrast to alkaline water electrolysis for hydrogen generation, PEM water electrolysis for hydrogen production employs the solid electrolyte of perfluorosulfonic acid proton exchange membrane with superior chemical stability, proton conductivity, and gas separation. PEM water electrolysis is characterized by its high efficiency, working current density, low volume, and high hydrogen purity [8,9]. Acidic electrolyte-based PEM water electrolyzers are superior to alkaline electrolyte-based water electrolyzers in terms of reduced resistance and increased current density, making them more suitable for practical application [10,11,12].

Hydrogen generation by PEM water electrolysis consumes a significant amount of energy due to the high overpotentials required for OER and HER. Since catalysts can reduce the activation energy of electrolytic water, thereby lowering its overpotential, the efficiency and cost of hydrogen generation from electrolytic water are highly dependent on the catalyst [13,14,15]. There has never been a break in the search for high-performance catalysts for water electrolysis. Currently, both precious metals, and non-precious metal-based compounds are employed as electrolytic water catalysts [16]. Catalysts made of non-precious metals have rapidly developed and shown excellent activity in water electrolysis. However, compared to platinum group metals (PGMs), these candidate materials are less stable under acidic conditions and their catalytic efficiency degrades quickly, hindering their industrial applications [17,18]. Because PEM becomes very acidic upon water absorption (equivalent to 10% H_2_SO_4_), only platinum group-based metals have potent activity against oxygen evolution reaction (OER) and hydrogen evolution reaction (HER) and can function reliably in severe environments. Currently, commercial OER catalysts are oxides of iridium (Ir) and ruthenium (Ru), while Pt/C is a HER catalyst. Platinum group metals (e.g., Pt, Ru, Rh, Ir, Pd, etc.) and their oxides have high catalytic activity for HER and OER and can effectively lower the overpotential of water electrolysis [19,20]. However, their wide application is limited by a lack of resources and high costs, which slows down the large-scale development of hydrogen production from electrolytic water. Therefore, it is necessary to maintain or enhance the efficiency of the catalysts while drastically lowering the loading of precious metals. Continuous development of innovative catalyst synthesis and electrode preparation technologies may be crucial in advancing the practical application of energy storage and conversion technologies [21].

In recent years, several cutting-edge techniques have enhanced the electrocatalytic performance of non-precious metal-based materials. However, there is still a gap between industrial requirements and the catalyst’s stability [22]. Due to this, the great focus should be placed on improving the electrocatalytic performance of noble metal-based catalysts. Transition metal composites [23], carbon-based supported materials [24,25], MXenes [26], metal-organic frameworks (MOFs) [27], conjugated polymers [28], and other materials interact synergistically with precious metals to aid in water electrolysis [29,30]. Noble metal-based materials, with their distinct nanostructure, pore structure, and small particle size nanocomposites, are regarded as some of the most important electrocatalyst materials due to their large specific surface area, surface active sites, and high catalytic activity [31,32]. 

This article provides an overview of current developments in the design and preparation of acidic OER and HER catalysts. We focus on precious metal-based catalysts for PEM water electrolysis, as opposed to prior evaluations that concentrated on a particular material or a few electrocatalytic processes. Initially, a short discussion of the mechanism for the reaction between acidic OER and HER is provided to facilitate a more thorough understanding of the EWS process and the development of more effective catalysts [33,34]. Here, we discuss the preparation methods based on noble metal catalysts and how they affect their catalytic performance, mainly including these Pt, Ru, Ir, Rh, and Pd-based catalysts. It is vital to notice that the beneficial impacts of precious metals are emphasized in the main text concerning active sites, conductivity, electronic structure, intermediate adsorption energy, etc. We also explain several modification techniques, including morphological design [35], interface engineering [36], defect engineering [28], etc. Finally, the difficulties and prospects associated with acidic water separation development are highlighted. All examples are essential suggestions for developing more effective OER and HER electrocatalysts for acidic environments. It is envisaged that these fundamental insights would aid in the quest for novel precious metal-based materials for increased usage in PEM electrolyzers. Thanks to the exhaustive investigations into precious metal-based catalysts for acidic EWS, researchers are expected to better understand the current state of research in the field, which will also inspire further endeavors. 

## 2. Progress in Catalysts for the OER

### 2.1. OER Mechanism

The OER is a multi-state reaction process, and the reaction kinetic is extremely sluggish, requiring a higher potential (1.23 V or higher) for the reaction, significantly reducing the energy conversion efficiency of the entire water decomposition [33,37,38]. Multistep synergistic and non-cooperative proton-electron transfer mechanisms may be used in the precise catalytic stages of OER [29,30]. The OER presents two different categories of catalytic processes: the adsorbate evolution mechanism (AEM) and the lattice-oxygen oxidation mechanism (LOM) [39]. More overpotential is necessary, which increases energy consumption. In addition, the OER is extremely pH sensitive, and the reaction mechanism varies depending on the electrolyte. In an acidic environment, two H_2_O molecules decompose into four protons (H^+^) and oxygen (O_2_). We provide the most well-known OER path below to aid comprehension of the OER process.

Under acidic conditions:(1)$$2H_{2}O+M\to M–OH+H_{2}O+H^{+}+e\;$$
(2)$$M–OH\to M–O+H^{+}+e\;$$
(3)$$M–O+H_{2}O\to M–OOH+H^{+}+e\;$$
(4)$$M–OOH\to O_{2}+H^{+}+e\;$$

According to the OER mechanism, the M-OH, M-O, and M-OOH intermediates may be formed in acidic environments. Advanced theoretical research also suggests that the performance of the OER catalyst is highly dependent on the speed determination stage; that is, the stages of the maximum energy barrier. The intermediate’s adsorption energy determines the catalytic reaction’s speed [40]. Therefore, catalysts with suitable energy barriers at each step will exhibit desirable OER activity. In comparison to HER, OER processes often include greater overpotential. OER significantly hampers the total energy efficiency of the electrochemical water splitting. Therefore, acquiring OER catalysts that can effectively minimize the theory barrier is vital.

Figure 1 depicts a typical AEM under acidic circumstances. The arrows in red show the responses of acid media. Firstly, M interacts with H_2_O to produce M-OH, which then loses an electron to generate M-O. O_2_ may be produced via two distinct processes: the first way involves the direct combination of two M-O to produce O_2_. Additionally, the second way involves the sequential reaction of H_2_O and M-O to produce M-OOH, followed by the production of O_2_ from M-OO. According to the Sabatier principle, the equilibrium between adsorption and desorption should be achieved [41]. The interaction between the catalyst and the reactive species during a catalytic reaction should not be too strong or too weak. If the effect is too strong, it is challenging to desorb the reaction product, and if the impact is too soft, it is challenging to combine the reactant species with the catalyst. Consequently, a modest binding capacity catalyst is required. From this process, if the adsorption link between O and the trigger is too low and the formation of OH is not easy, it can be deduced that OOH will be formed. The peak performance of the catalyst can only be achieved when O and the catalyst liaison capacity are not intense under average conditions [42].

### 2.2. Advanced Electrocatalysts for Acidic OER

Multiple responses with four electrons motions are pretty tricky. Due to the complexity of the OER, the theoretical reaction speed is slowed down and energy efficiency is significantly diminished. 

**Table 1 nanomaterials-12-02618-t001:** Summary of the OER performance for the electrocatalysts in an acidic medium.

Catalyst	Electrolyte	Activity at 10 mA/cm^2^ (mV)	Stability	Refs.
IrCo@IrCoO_x_	0.5 M H_2_SO_4_	259	55 h@50 mA/cm^2^	[43]
IrNiO_x_/C-A	0.05 M H_2_SO_4_	400	–	[44]
IrCuNi	0.1 M HClO_4_	273	1800 cycles	[45]
IrO_x_-networks	0.1 M HClO_4_	300	12 h@1.6 V_RHE_	[46]
Ru NCs/Co_2_P HMs	0.5 M H_2_SO_4_	197	12 h@161 mA/cm^2^	[47]
IrRuMn	0.1 M HClO_4_	260	8 h@10 mA/cm^2^	[48]
Ir@N-G-600	0.5 M H_2_SO_4_	314.6	28 h@15 mA/cm^2^	[49]
Cr_0.6_Ru_0.4_O_2_	0.5 M H_2_SO_4_	178	10 h@10 mA/cm^2^	[50]
Cr_0.6_Ru_0.4_O_x_-350	0.5 M H_2_SO_4_	250	25 h@10 mA/cm^2^	[51]
Ir_3_CeO_x_/C	0.5 M H_2_SO_4_	299	10 h@10 mA/cm^2^	[52]
RuCu NSs/C-350℃	0.5 M H_2_SO_4_	236	12 h@10 mA/cm^2^	[53]
Macro-RuO_2_	0.1 M HClO_4_	263	2 h@10 mA/cm^2^	[54]
RuO_2_@IrO_x_	0.5 M H_2_SO_4_	215	300 h@1 A/cm^2^	[55]
Ru-RuO_2_@NPC	0.5 M H_2_SO_4_	192	–	[56]
TiO_2_-RuO_2_	0.1 M H_2_SO_4_	386	6 h@1 mA/cm^2^	[57]
Ru-N-C	0.5 M H_2_SO_4_	267	30 h@10 mA/cm^2^	[58]
IrO_2_ (1:100)-450 °C	0.5 M H_2_SO_4_	282	–	[59]
Ir_6_Ag_9_ NTs	0.5 M H_2_SO_4_	285	6 h@5 mA/cm^2^	[60]
E-Ru/Fe ONAs	0.5 M H_2_SO_4_	238	9 h@5 mA/cm^2^	[61]
Ru_3_Ni_3_ NAs	0.5 M H_2_SO_4_	268	10 h@5 mA/cm^2^	[62]
0.5Ru_0.1_Cu-GN1000	0.5 M H_2_SO_4_	157	1000 cycles	[63]
30Ir/Au/CP	0.5 M H_2_SO_4_	318	8 h@10 mA/cm^2^	[64]
Au-Ir	0.1 M HClO_4_	351	5 h@10 mA/cm^2^	[65]
RuO_2_-Ru/GDY	0.5 M H_2_SO_4_	163	75 h@10 mA/cm^2^	[66]
Ir–Te NWs	0.5 M H_2_SO_4_	284	20 h@5 mA/cm^2^	[67]
Ir-NS	0.5 M H_2_SO_4_	254	50 h@10 mA/cm^2^	[68]
Ag_1_/IrO_x_	0.5 M H_2_SO_4_	224	50 h@10 mA/cm^2^	[69]

Therefore, practical, highly active, and long-lasting catalysts with stable electrodes are required to overcome the considerable energy barriers [40,70]. Until now, most electrocatalysts for acidic OER have been based on Ir and Ru nanostructures, such as monometallic and oxides. Most newly developed precious metal-based compounds display more OER activity and stability than commercially available IrO_2_ and RuO_2_, hence reducing the need for costly Ir and Ru. 

Table 1 summarizes the current research development of these sophisticated Ru- and Ir-based electrocatalysts for improving acidic OER.

#### 2.2.1. Ru-Based Electrocatalysts

RuO_2_ has the most excellent OER activity compared to other OER catalyst materials. However, it is vulnerable to severe corrosion while operating in acidic electrolytes. During the OER process, it has been reported that oxygen molecules are generated from the lattice oxygen of RuO_2_, resulting in enhanced instability and considerable depletion of the Ru element in the acidic electrolyte [71,72]. To improve their OER performance in acidic circumstances, research has been undertaken on Ru and its associated materials. Specifically, it is essential to extend its durability while maintaining its high catalytic activity.

It is standard practice to design Ru-based catalysts with nanostructures and variable porosity. The objective is to enhance the catalytic activity by increasing the surface area and exposing the active spots. Changing the morphology and composition of Ru nanosheets, for instance, may significantly boost their OER catalytic performance. Qing et al. designed a RuCu nanosheet rich in channels as an effective electrode catalyst for OER (Figure 2a,e) [53]. The thickness is approximately 6 nm and their RuCu NSs are observed with highly accessed channels in the TEM image (Figure 2d,e). From a structural point of view, the two-dimensional (2D) structure offers a large surface and improves electrochemical performance by exhibiting more atoms as active points. As shown in Figure 2c, the optimized sample in 0.5 M H_2_SO_4_ can supply 10 mA cm^−2^ at 1.49 V. DFT simulations (Figure 2f) revealed that in the initial state (U = 1.23 V), water adsorption and decomposition are unimpeded. Then, a 0.164 eV energy barrier is shown for the OER process. RuCu NSs have excellent electrocatalytic activity, owing primarily to the combination of lattice deformation in the channel area and an optimized electronic structure. More active sites may be exposed by producing nanostructures (nanotubes, nanosheets, etc.) of Ru-based catalysts.

OER activity is dependent on the connection between the electronic structure of the active metal and the adsorption/desorption characteristics of the reaction intermediate. The electronic structure of Ru atoms can be altered by doping with heteroatoms. The OER activity of the catalyst may be significantly enhanced by speeding the dissociation of water and lowering the reaction intermediate adsorption and reaction energy barriers [73]. Prior research has shown that the inclusion of transition metal atoms, such as Co and Ni, into the Ru lattice to construct multi-alloys can enhance the charge distribution and surface structure of the catalyst, hence improving its catalytic activity. In addition, doping with metal ions, particularly transition metal ions, can reduce ruthenium loading with relative ease. For example, Yao et al. inserted Pt elements into the Ru oxide lattice to generate Pt-Ru nanosheet assemblies that encouraged the creation of oxygen vacancies with more excellent OER activity than pure ruthenium oxide nanosheets [74]. Huang et al. increased the acidic OER activity of ruthenium-based catalysts by including Pd elements to create RuPbO_x_ [75]. Yang and colleagues reported a three-dimensional (3D) hierarchical assembly structure with a Ru-Ni alloy core, applied to the OER procedure. Nanoflowers with a three-dimensional structure that is stacked on two-dimensional nanosheets were fabricated. Compared to a typical Ir/C, this reported system exhibited excellent catalytic characteristics and sustainable characteristics under an acidic medium. From the polarization curves, Ru_3_Ni_3_ NAs displayed the highest OER activity in 0.5 and 0.05 M H_2_SO_4_ (Figure 3a,b). To drive a current density of 10 mA cm^−2^, the Ru_3_Ni_3_ NAs, Ru_3_Ni_2_ NAs, and Ru_3_Ni_1_ NAs required overpotentials of 252, 260, and 268 mV, respectively. Ru_3_Ni_3_ NAs exhibited minimal overpotential after 10 h of steady cycling in an acidic environment. (Figure 3e,f). Lattice doping may improve the electrochemical structure of the substrate material by exposing additional active sites while simultaneously maintaining the system. Adding Ni decreases the d-band center of Ru_3_Ni_3_ NAs and changes the surface electrical condition. Ru enhances the electron transfer between the catalytic base and middle molecules, promoting the production of O-O bonds, according to DFT simulations. Consequently, the Ru-Ni (NAs) system has an excellent capability for EWS.

As shown in Figure 4a, Huang’s research group demonstrated the chemical engraving strategy to prepare Ru/Fe oxides by HNO_3_, and in the original ingredient of Ru/Fe (P–Ru/Fe NAs). The P–Ru/Fe NAs were partially etched by nitric acid (HNO_3_), resulting in a certain number of holes in the nano component of Ru/Fe engraved (E-Ru/Fe ONAs). The acquired samples are rich in ultrathin subunits (Figure 4b,c). During the OER process, the ruthenium oxide lattice makes ruthenium atoms water-soluble and bond-formation possible. As the iron is etched away, the number of electrons in the oxygen lattice connected to the Ru atom increases. This makes water preferentially stick to the metal atom. Vacancy flaws may also improve the conductivity of the catalyst. Grain boundaries and dislocations are readily apparent in the HRTEM picture (Figure 4h,i). Consequently, the optimized E-Ru/Fe ONAs exhibit superior OER activity with a low overpotential of 238 mV at 10 mA cm^−2^ in 0.5 M H_2_SO_4_. According to DFT simulations, a significant number of vacancies at the interface of Ru-Fe oxides may not only alter the electronic state in the O lattice and inhibit the dissolution of RuO_2_, but also improve OER activity by increasing the binding capacity of the intermediates. It is also feasible for this research to demonstrate an effective way of stabilizing Ru-based catalysts.

#### 2.2.2. Ir-Based Electrocatalysts

Ir is a precious metal that is 10 times scarcer than platinum. Consequently, Ir’s rarity and high cost hinder its wide range of applications as a catalyst. Despite this, Ir and related oxides as OER catalysts have an excellent balance between catalytic activity and stability [76]. In 1978, scientists found that Ir-based materials were the best OER catalysts in acidic environments [77]. Now, Ir or IrO_2_ is the central active part of anode catalysts. The OER activity of IrO_2_ is slightly smaller than that of RuO_2_ under acidic circumstances, while the lifespan of IrO_2_ is twenty times greater than that of RuO_2_. Therefore, selecting suitable doping or carrier materials may significantly minimize Ir loading and enhance the corrosion resistance of the electrode. A typical technique for reducing IrO_2_ load is to add cheap metals that can change their electrical and structural properties, leading to improved activity and stability. An IrW nano-channel has been created as a carrier for IrO_2_, which modifies the charge distribution of Ir atoms on the surface and inhibits Ir from collecting additional oxygen, resulting in a super-stable OER electrocatalyst [78]. The electrode catalyst obtained is a binary or ternary complex of Ir_x_M_y_N_z_O_a_ shapes and other metal oxides where M and N are inexpensive metals. For instance, Li et al. fabricated a nanoarray porous catalyst (Ir_x_Ru_1-x_O_2_) that displayed significant intrinsic activity.

Heterostructure catalysts are often composed of two or more materials that are physically or chemically bound. Creating heterostructure OER catalysts is an efficient method for increasing the number of active sites, primarily by employing nanostructures with clearly exposed edges that offer adequate adsorption sites for OER intermediates. Consequently, their OER activity is greater than that of single-material catalysts. According to the research of Chen and colleagues, Ir can increase the OER activity by establishing phase boundaries with Au. The OER-induced structural evolution of the carbon paper electrode leads to the formation of a catalyst with an Au-Ir-rich interface, as shown in Figure 5c,f. Nanostructured catalysts may be manufactured on a base material that serves as a support to prevent agglomeration of the active component. Compared to the Ir catalyst, which requires around 393 mV to obtain 10 mA/cm^2^, the overpotential of the evolved Au-Ir catalyst is only 351 mV (Figure 5d,e). Ir has significant binding energy to O-species, making it easy to break down H_2_O but not conducive to the production of O_2_. To address this deficiency, Au was chosen as another component of the composite catalyst owing to its poor binding energy to O-species, which might also help oxygen synthesis during the intermediate stage of OER. Since the beginning potentials are almost equal, this shows that the Ir active sites on the samples commence the OER process and that Au has little influence on the initial activity of Ir locations. Catalysts for the OER process go through structural change, particle development, and an increase in the Au-Ir phase boundary. Therefore, heterostructure Au-Ir catalysts have a higher OER activity than their monometallic counterparts. A great working potential for more oxygen-containing intermediates to accumulate on the Ir sites may subsequently be transferred to the Au sites to produce oxygen, resulting in a higher current density than the Ir catalyst alone delivers. We anticipate that combining Ir-based nanoparticles with Au to build heterogeneous structures, such as IrCo, IrNi, and IrCu, may provide more engaging OER catalysts (Figure 5i).

**Figure 5 nanomaterials-12-02618-f005:**
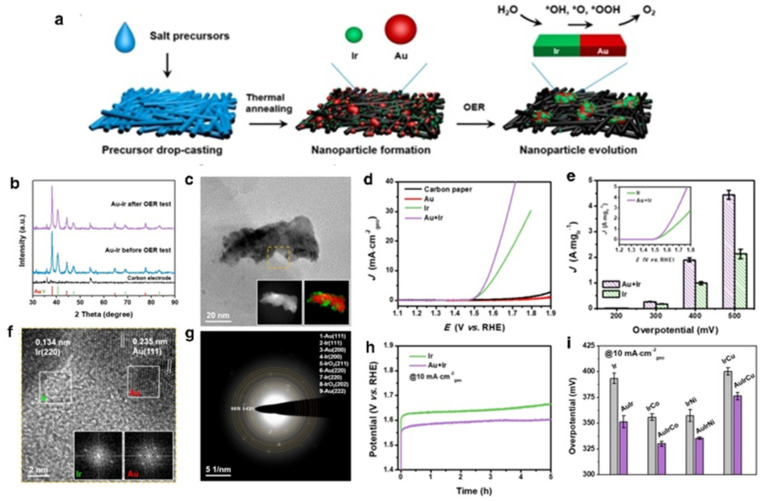
(**a**) Scheme demonstrating the fabrication of Au–Ir catalysts, (**b**) XRD patterns of carbon paper and Au–Ir catalysts, (**c**) TEM image of a representative Au-Ir heterostructured particle formed after OER test, insets are HAADF–STEM image and EDS mapping of the particle, (**d**) LSV curves of different catalysts in 0.1 M HClO_4_ solution, (**e**) Ir–mass–based OER activities of Ir and Au-Ir catalysts, (**f**) HRTEM image of the region indicated by a yellow dashed square in the particle shown in (**c**), (**g**) Electron diffraction pattern of the particle shown in (**c**), (**h**) Chronopotentiometric measurements of Ir and Au-Ir catalyst, (**i**) Overpotentials of different Ir-based motivations at 10 mA/cm^2^. Reprinted with permission from Ref. [65], Copyright 2021, *American Chemical Society*.

**Figure 6 nanomaterials-12-02618-f006:**
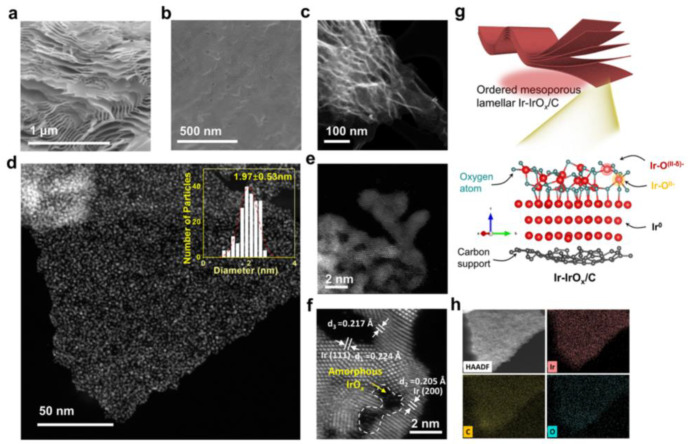
Characterization of the Ir–IrO_x_/C-20. (**a**) surface image, (**b**) cross–section SE, (**c**,**d**) Low-magnification HAADF–STEM image. Inset in (**d**): The nanoparticle size statistics diagram, (**e**,**f**) Spherical aberration–corrected high-resolution AC–HAADF–STEM images, (**g**) Illustration of the structure. (**h**) STEM–EDS element mapping images. Reprinted with permission from Ref. [79], Copyright 2022, *American Chemical Society*.

The electronic structure of two-dimensional (2D) stacked catalysts is varied. These materials possess a high specific surface area and a tunable interlayer space. The oxidizing status of the surface may also change in addition to the increase in active sites [80]. To optimize the usage of active sites, it is required to develop an efficient approach for synthesizing Ir-based two-dimensional electrocatalysts with ordered/tunable interlayer gaps. Self-assembly is shown to be a versatile synthetic approach for synthesizing catalysts with adjustable ordered porous architectures and evenly dispersed compositions. Recent research has shown that stable end-fitted sheet micelles may be in nanoconfined self-assembly processes that can produce thin Ir-IrO_x_/C nanosheets with regulated interlayer gaps. According to SEM images, the Ir-IrO_x_/C-20 catalyst exhibits a consistent two-dimensional layered structure with clearly visible organized mesopores and nanochannels. (Figure 6a,b). The interlayer spacing of Ir-IrO_x_/C nanosheets was carefully tailored to 20 nm, and Ir-IrO_x_ nanoparticles (~2 nm) were distributed equally across the nanosheets (Figure 6c,d). Significantly, when treated to OER in acidic environments, the produced Ir-IrO_x_/C electrocatalysts displayed the overpotentials of 198 mV at 10 mA cm^−2^. Due to their mixed valence, an abundance of electrophilic oxygen species, and advantageous spatial organization, metallic Ir nanocrystals increase the uptake capacity of the oxygen molecule. High specific surface area and open interlayer channels offer more active sites and boost ion and electron transport efficiency. The unique oxygen coordination environment renders water molecules more susceptible to nucleophilic attack and accelerates the creation of O-O bonds. Furthermore, numerous works use MOFs derivatives as carriers for iridium oxide. For instance, Xu et al. constructed a structure in which RuIr nanocrystals are evenly disseminated on a carbon carrier formed from a metal-organic framework skeleton (CoNC). The noble metal atoms have strong interactions with the carriers, and the synergistic effect of Ru and Ir considerably improves the OER performance [27]. In addition, OER electrocatalysts frequently employ MOFs and their derivatives because of their many pore architectures, substantial specific surface area, varied compositions, and clearly defined metal centers [81]. However, some significant problems and difficulties need to be handled. Even though thousands of MOFs have been reported, the majority have poor response stability, which greatly restricts their scale-up applications. Novel design approaches are thus required to increase the MOFs’ accessibility. Intrinsic conductivity is a significant obstacle to the practical use of MOFs. Uncertainty surrounds the catalytic mechanism of MOFs for OER. The electrocatalytic activity of MOFs for OER is often carried out in alkaline fluids due to the quick reaction kinetics [82].

## 3. Progress in Catalysts for the HER

### 3.1. HER Mechanism

HER is a basic two-electron transfer process. The corresponding procedure may be described in broad strokes. First, the H ion forms H*, which is adsorbed in the active portion of the catalyst. Then, H* is bound in a particular manner, and H_2_ is produced as a result [83,84].

HER, like OER, has been discovered to be particularly sensitive to pH. One hydrogen molecule may be produced in acidic solutions, but in neutral and basic conditions, two H_2_O molecules can be converted to H_2_ and -OH. The Volmer–Heyrovsky and Volmer–Tafel mechanisms are likely the best explanations for the HER mechanism based on the existing research (Figure 7). The various processes involved in the reaction are listed below:

In the acidic media:(5)$$H^{+}+e\to M–H\left(Volmer\;step\right)$$
(6)$$M–H+M–H\to H_{2}\left(Tafel\;step\right)\;$$
(7)$$H^{+}+e+M–H\to H_{2}\left(Heyrovsky\;step\right)\;$$

**Figure 7 nanomaterials-12-02618-f007:**
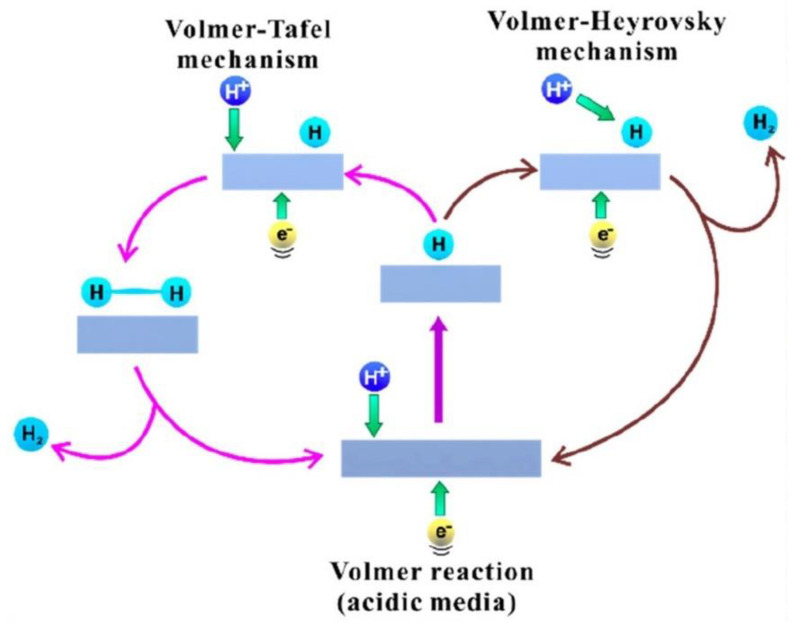
Mechanism of HER action on electrode surfaces in acidic media. Reprinted with permission from Ref. [85], Copyright 2019, *American Chemical Society*.

According to the HER process, the HER performance of catalysts depends on the free energy of adsorbed hydrogen atoms. Hence, materials with thermoneutral qualities can possess high HER activity.

### 3.2. Advanced Electrocatalysts for Acidic HER

Due to their near-zero Gibbs free energy of hydrogen adsorption, platinum group metals are currently regarded as the best catalysts for HER. We thoroughly analyzed these most recent developments in acidic OER enhancement electrocatalysts based on platinum group metals, the performance of which is described in Table 2. Ru-based catalysts may boost their catalytic activity by exposing additional active sites through structural design, conductive carrier support, electronic structure tuning, and interfacial structure design.

#### 3.2.1. Pt-Based Electrocatalysts

Platinum (Pt) is the well-recognized HER catalyst since there is almost no overpotential at startup and the current rises rapidly with increasing voltage. Reducing the loading of platinum is the only viable method for reducing prices, and researchers have made several efforts to build new catalysts to develop inexpensive and plentiful alternatives to Pt [109,110]. Although it is commonly asserted that potential non-Pt catalysts have been developed, none can match the inherent activity and stability of Pt/C. Therefore, creating new Pt-based metallic electrocatalysts is particularly important.

Due to the unique electrical structure of Pt, the binding strength between the active site and the hydrogen ion is optimal. Creating heterostructured catalysts made of electrochemically active materials and other functional additions can improve catalytic performance during HER while reducing Pt loading. Sun et al. presented a novel use of Co/NC heterojunctions as “solid ligands” for stabilizing and activating ultrafine Pt nanoparticles to achieve effective HER activity. Co/NC heterojunctions could regulate the generation of ultrafine Pt nanoparticles in high concentrations (Figure 8a,b). The strongly linked interface between Co/NC and Pt boosts the electron density of Pt nanoparticles, enhances the adsorption capacity for hydrogen ions, and significantly decreases the overpotential of Pt catalysts. At the same time, the ultra-high mass activity makes it applicable in the real world. This research improved the activity of the catalyst by altering the ratio of electron acceptor (Pt) to the electron donor (Co/NC), resulting in an 8.3-fold increase in the HER mass activity of Pt_x_/Co compared to commercially available Pt/C. When 10 mA cm^−2^ was supplied with the addition of platinum nanoparticles, the overpotential was reduced from 285.1 mA at the Co/NC electrode to 6.8 mA at the Pt_4_/Co electrode, as predicted (Figure 8e,f). Figure 8g and h demonstrate that Pt_4_/Co-based catalysts may fulfill the high-performance electrode durability and activity criteria.

The design and synthesis of materials provide a viable method for bit-efficient HER. Ma et al. reported a heterogeneous structure of Pt and vanadium carbide (V_8_C_7_), wherein V_8_C_7_ as a carrier can modulate the electrical structure of Pt nanoparticles and H adsorption on the surface to improve metal atom utilization, enabling the sample to exhibit ultra-high activity under acidic conditions [111].

#### 3.2.2. Ru-Based Electrocatalysts

Non-precious metal-based and metal-free catalysts have been developed as possible replacements for platinum-based catalysts during the last few decades. In PEM water electrolysis, however, their high overpotential and limited durability make them challenging to use. Ru can potentially replace Pt as an acidic HER catalyst since it is just one-fifth as expensive as Pt metal [34,112,113]. This section discusses the function that Ru nanoparticles (Ru NP) support in HER activity. When Ru NP is connected to various substrates, including carbon, carbonaceous composites, metal, and semimetal-based materials, their HER performance is virtually always enhanced [114,115].

MOFs comprised of transition metal ions and organic ligands have been extensively exploited as building blocks for the creation of functional nanomaterials in recent years. This is because their structure, pore size, and distribution of metal sites can be tailored to specific requirements [116]. Due to the large number of active sites exposed in the pore channels and the weak coordination bonds between the metal sites and organic linkers, which are vulnerable to the electrochemical environment, primitive MOFs are seldom utilized directly as catalysts. MOFs with rapid electron transport and increased specific surface area in two-dimensional structures are being explored as possible catalyst carriers. Lin and his coworkers fabricated two-dimensional MIL-53 (NiFe) MOF nanosheets supported by ultralow Ru-doped bimetallic phosphides (Ru-NiFeP/NF) on nickel foam. (Figure 9a). The NF substrate has a reasonably smooth surface and a three-dimensional porous structure, as shown in Figure 9b. Ru-Ni_2_P/NF created a three-dimensional open porous structure with vertically oriented nanosheet arrays, as illustrated in Figure 9c. Ru-FeP was mainly formed as tiny irregularly shaped particles, as seen by SEM pictures (Figure 9d). The nanosheet structure retained its original shape after phosphorylation; however, the Ru-NiFeP produced had a coarser texture (Figure 9e). EDX analysis shows that Ru, Ni, Fe, P, C, and O atoms are evenly distributed throughout the sample, further supporting the effective inclusion of Ru atoms (Figure 9f). Under acidic circumstances (Figure 9g), NiFeP/NF with an optimized electrical structure exhibits good HER performance, needing just 29 mV of overpotential to produce 10 mA cm^−2^. Despite Ru’s superior characteristics, the study of Ru-based catalysts for acidic HER is still in its infancy. Therefore, it is necessary to continually investigate systematic strategies for designing and producing Ru-based catalysts. Other researchers have generated nano-RuW composite catalysts via magnetron sputtering, where the alloying enlarges the cell and electrons are transported from W to Ru atoms, thus altering the electron structure of ruthenium and increasing its electrocatalytic activity [117].

**Figure 9 nanomaterials-12-02618-f009:**
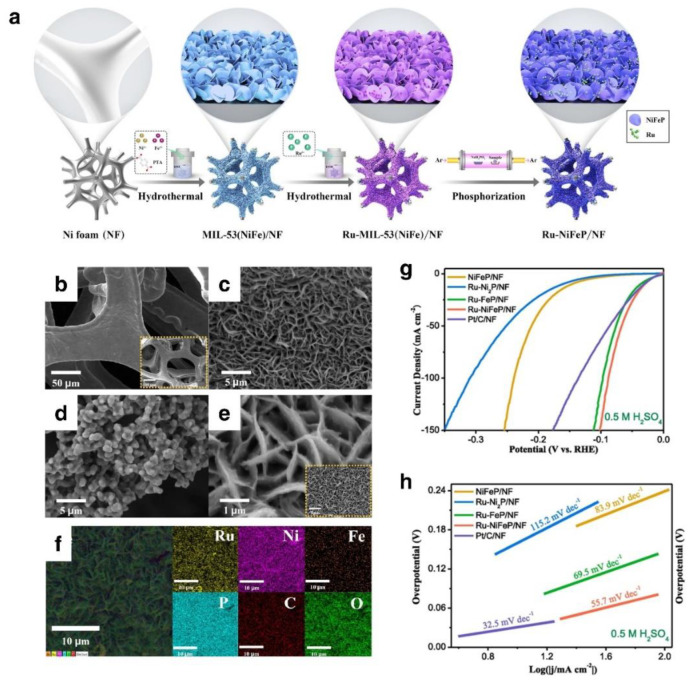
(**a**) Schematic diagram of the synthetic route of Ru–NiFeP/NF nanosheets, SEM images of (**b**) bare NF, (**c**) Ru–Ni_2_P/NF, (**d**) Ru–FeP/NF, (**e**) Ru–NiFeP/NF, (**f**) EDX elemental mapping of Ru–NiFeP/NF, (**g**) LSV curves recorded in 0.5 M H_2_SO_4_, (**h**) Corresponding Tafel plot. Reprinted with permission from Ref. [102], Copyright 2021, *Elsevier*.

**Figure 10 nanomaterials-12-02618-f010:**
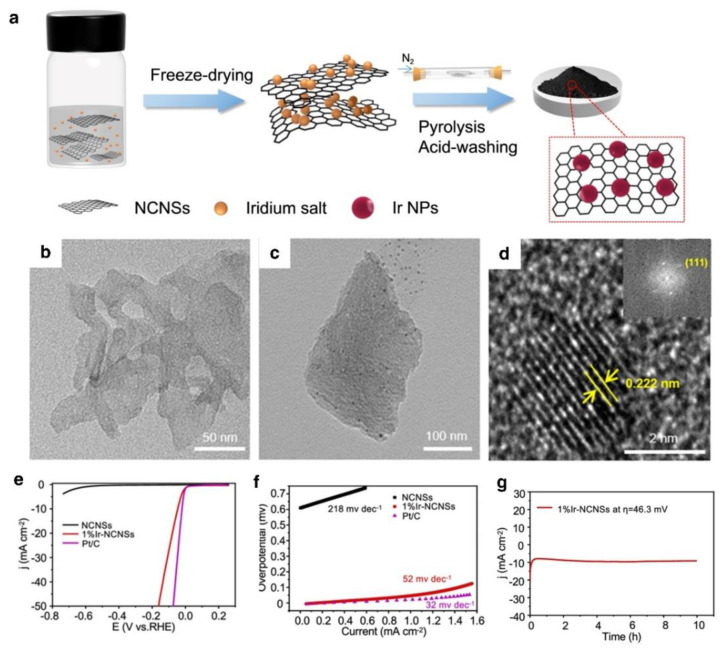
(**a**) Synthesis route of Ir–NCNSs catalyst, (**b**,**c**) TEM image of the NCNSs and 1%Ir–NCNSs, Characterization of 1%Ir–NCNSs: (**d**) High-resolution TEM image, (**e**) Polarization curves, (**f**) Tafel plot, (**g**) Chronoamperometry curve. Reprinted with permission from Ref. [118], Copyright 2021, *American Chemical Society*.

#### 3.2.3. Ir-Based Electrocatalysts

Ir is located next to Pt on the volcano diagram, which is an excellent option for acidic HER catalysts. Two-dimensional nanosheets have a high specific surface area and anisotropy, which expose more active sites and generate more defects on the surface, resulting in an abundance of nucleation sites for the development of metals and distinct benefits in the HER process [119,120]. According to Dai’s research, Iridium nanoparticles (Ir NPs) were spontaneously deposited on a 2D siloxane substrate. At ultra-low Ir loading, the resultant Ir NPs/siloxane catalysts displayed outstanding HER electrocatalytic activity [121]. Two-dimensional siloxanes act as efficient carriers for highly dispersed Ir NPs with highly reductive and anchorable functional groups on the surface, allowing ultrafine Ir NPs to be fabricated spontaneously. With 10 mA cm^−2^ at an overpotential of 31 mV, the Ir NPs/siloxane catalyst had an Ir loading of 2.1 wt %, and its activity had already surpassed that of commercial platinum carbon material. According to this work, catalytic activity can be improved by modifying the electronic structure of the metal–carrier interaction. In contrast, two-dimensional layered siloxanes are excellent catalyst carriers for energy conversion and storage applications. In addition, Wu et al. presented an Ir-based HER catalyst supported by N-doped carbon nanosheets (Ir–NCNSs) (Figure 10a). The 2D N-doped carbon nanosheets (NCNSs), with a large surface area and distinctive atomic structure, enable Ir NPs to disperse at 2–3 nm and strongly coordinate with the Ir through Ir–N bonds at numerous active sites, strengthening their endurance (Figure 10b). Due to the anchoring strategy, Ir NPs are evenly distributed in the NCNS matrix (Figure 10c) and Ir crystal is well demonstrated in Figure 10d. Compared to the pure NCNS, 1%Ir–NCNSs show a very modest overpotential of 46.3 mV at 10 mA cm^−2^ (Figure 10e) and a low Tafel slope (Figure 10f). Generally, the primary active sites are Ir NPs implanted in the N6 hollow area. As a result, the N6 hollow sites contribute to the improvement of HER characteristics and the anchoring of Ir NPs.

Utilizing catalyst supports, which aid in improving dispersion and decreasing agglomeration of active electrocatalysts, is another strategy for reducing catalyst loading [122,123]. Due to its vast surface area, high electrical conductivity, variable graphitization, and pore structure, carbon are generally the most well-known and most often used support material in electrochemistry [124]. Li et al. created an Ir@S-C/rGO catalyst with nanodots supported by reduced graphene oxide [125]. Wu and colleagues present an iridium nanoparticle with high activity dispersed on nitrogen-doped graphene sheets [126]. These examples demonstrate how appropriate supports, mainly carbon, may work in concert with catalysts to control hydrogen, while also possessing HER activity on their own. Electrocatalysts’ internal resistance can be decreased by combining them with highly conductive carbon substrates [122,127]. Additionally, nanoparticles are vulnerable to corrosion in the water electrolyzer’s operating environment. In this instance, the carbon support improves the nanoparticles’ resilience in corrosive settings. Some people have even proposed a very active direct synthesis of HER catalysts with Ni and N co-doping from CO_2_. A clean transition from a carbon-based economy to a hydrogen-based economy is achieved by comparing the physical qualities and electrocatalytic activities of carbon compounds generated from CO_2_ [128,129].

**Figure 11 nanomaterials-12-02618-f011:**
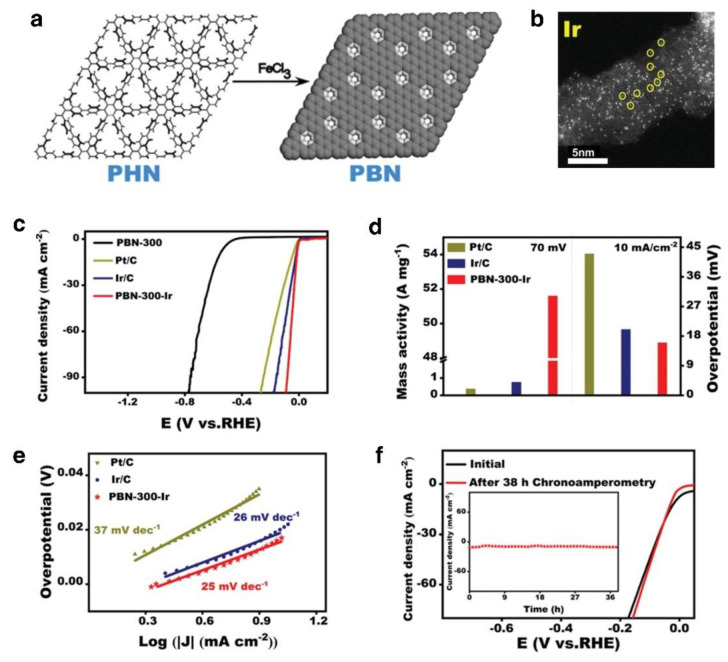
(**a**) Synthesis route of PBN, (**b**)ACHAADF–STEM imaging of PBN-300-Ir, (**c**) LSV curves for these catalysts in 0.5 M H_2_SO, (**d**) Comparison between the mass activity at 70 mV and the overpotentials required to achieve 10 mA cm^−2^ (**e**) Tafel plots, (**f**) Comparison of LSV curves before (black) and after (red) chronoamperometry test, The inset is current density versus time (I–T) curves of PBN-300-Ir recorded for 38 h at −0.019 V versus RHE. Reprinted with permission from Ref. [130], Copyright 2022, *John Wiley and Sons*.

To improve the catalytic activity of HER, Ir-based electrocatalysts containing nanostructures such as nanowires, nanosheets, nanotubes, nano dendrites, nano frames, and single atoms have been intensively studied [108,131,132]. Xie et al. synthesized ultrathin Ir nanosheets (~4 nm) using MIL-88A as a sacrifice template. At an overpotential of 50 mV, the HER catalytic performance of Ir nanosheets was 5.5 times higher than that of regular Pt/C. Ir NS appears to be more connected with oxidized Ir species than Ir NP, which might be due to higher surface atom exposure. This is because the outermost Ir atoms are more likely to be oxidized, resulting in an enhanced Ir valence state. Furthermore, the presence of oxygen atoms alters the electronic structure of Ir atoms by decreasing the d-band center, which is an effective technique to increase Ir’s HER characteristics. Meanwhile, the Ir NS electrocatalyst shows the excellent electron/ion conductivity of the nanosheet structure, which is helpful for the formation of a rapid electron transport channel.

Numerous research teams have been developing supported Ir single-atom catalysts (SACs). For instance, the holey carbon skeleton (PBN) was prepared from the polyhexaphenylbenzene network (PHN), as shown in Figure 11a [130]. The Ir ions that have been ligated are reduced to single atoms and incorporated into the support PBN (PBN-300-Ir), with an Ir element concentration of approximately 0.74 percent (Figure 11b). When obtained at 10 mA cm^−2^ in 0.5 M H_2_SO_4_, PBN-300-Ir had an overpotential of just 17 mV. This suggests that PBN-300-Ir has a very high Ir atom utilization efficiency (Figure 11c,d). After 38 h of HER reaction in a highly acidic electrolyte, the current density of PBN-300-Ir remained almost unaltered in Figure 11f, demonstrating that PBN-300-Ir has high stability. Most of the published research on precious metal-based single-atom catalysts (PMSACs) focuses on single atoms such as Ir, Ru, Pt, Rh, and Pd anchored on diverse conductive substrates. With its maximum atom use and unique electronic structure, PMSAC has the potential to contribute significantly to the advancement of PEM water electrolysis technology by preserving good catalytic performance while using less metal. Further improvement of the loading density of successfully exposed single-atom sites in PMSACs is necessary to achieve high specific mass/area activity and offer a plentiful supply of active sites for electrocatalysis.

#### 3.2.4. Other Electrocatalysts

It has been shown that PGMs, such as Pt, is the most effective HER electrocatalysts. To decrease the consumption of PGMs with improved utilization efficiency of PGMs, the size and nanostructure of PGM-based electrocatalysts must be accurately regulated [133]. Pd is suggested as one of the substitutes for HER electrocatalysts because of its near closeness to Pt in the volcano diagram with hydrogen absorption energy. Pd is considered a viable substitute for Pt, and several ways have been devised to boost its electrocatalytic activity [84]. By mixing Pd with transition metal M (e.g., Fe, Co, Cu, etc.), the existed synergistic and ligand effects, can result in the alternation of electronic structure and adsorption free energy, leading to the improved performance of HER. In addition, the formation of Pd hydride (PdH_x_) diminishes the adsorption of hydrogen on the catalyst’s surface and enhances hydrogen generation. However, the stability of Pd-based HER catalysts remains challenging. Jia et al. synthesized a stable Pd-Cu hydride (PdCu_0.2_H_0.43_) catalyst that combines the benefits of Pd m and PdH_x_ structures and considerably improved the endurance in acidic HER environments (Figure 12a). A stable Pd-based hydride catalyst is produced using a novel synthetic approach that permits hydrogen doping at atmospheric pressure. The derived PdCu_0.2_H_0.43_ catalyst has very low overpotential and Tafel slope values (Figure 12b,d). Due to the considerable free energy of hydrogen absorption, it also demonstrates exceptional durability. According to theoretical estimation, the enhancement of the hydrogen adsorption capacity on the PdCu_0.2_H_0.43_ structure is mainly attributed to the doping of the less reactive Cu and the repulsion with interstitial hydrogen atoms to weaken the H* bond.

Metal nanoclusters (MNCs) have gained substantial attention in electrocatalysis owing to their unusual electrical structure and high surface atomic dispersion. However, owing to their high surface energy, MNCs move and aggregate rapidly during the catalytic process, reducing catalytic activity and restricting their utilization. Surface capping agents may be used to regulate the nanostructure and size of MNCs. According to Ding et al., the manufacture of well-distributed, monodisperse metal NCs (Rh, Ru, and Ir) has been reported [134]. They employed NHCSs (nitrogen-doped hollow carbon spheres) as support carriers. MNCs with diameters smaller than 2 nm were consistently dispersed on the surface of NHCSs owing to the confinement of carbon nanopores and the anchoring action of nitrogen atoms. Hierarchical porous designs in NHCS substrates can provide many transport channels for ion diffusion and electron transfer. Rh/NHCSs are exhibiting promising HER electrocatalytic activity (10 mV in 0.5 M H_2_SO_4_). The O and N-dopants in NHCSs are expected to attract the attention of NP precursors and serve as anchor sites for the nucleation and development of metal nanoparticles, thereby stabilizing the NPs with smaller and more uniform sizes. Furthermore, the high electrostatic adsorption of these functional groups to the metal particles also guarantees that the MNCs are securely attached to the NHCSs, which is another benefit. Yao and coworkers have prepared 3D Rh/Rh_2_P nano-flake assembly (Rh/Rh_2_P-NFAs) using partial phosphating (Figure 13a) [135]. High-resolution TEM pictures (Figure 13b) clearly show the two different lattice spacings of Rh and Rh_2_P. This indicates the successful generation of the Rh-Rh_2_P heterostructure. In an acidic medium, Rh/Rh_2_P-NFAs require only a 13.4 mV overpotential to achieve 10 mA cm^−2^ for HER (Figure 13c), which is less than that of Rh-NFAs (40.6 mV) and Pt/C (27.1 mV). The Tafel slope values and charge transfer resistance (Rct) of Rh/Rh_2_P-NFAs were lower than those of Rh-NFAs and Pt/C (Figure 13d,e), indicating that the electron transfer ability was accelerated during HER due to the efficient binding of P that enhances proton adsorption and desorption, resulting in a large Rh-Rh_2_P heterogeneous surface. Furthermore, Rh/Rh_2_P-NFAs have demonstrated long-term stability. More attention is being paid to Rh-based catalysts, which are regarded as promising and viable HER candidates.

## 4. Conclusions and Perspectives

In terms of resource consumption, environmental protection, and effective use of hydrogen energy, the EWS has vast development potential. In summary, we provide an overview of recent advances in studying precious metal-based electrode catalysts for EWS in acidic conditions, mainly containing precious metal materials (e.g., Ir, Ru, Pt, Rh, and Pd). First, the mechanism of acid electrolysis of water, particularly the reaction steps in OER and HER processes, is discussed in detail. Then, we summarized the recent research advancements of precious metal-based electrocatalysts in acidic OER and HER. The performance of the reported catalysts is compared in the PEM water electrolysis processes. Meanwhile, the design principles and preparation process of catalysts based on precious metals were described (e.g., structural design, interface engineering, defect engineering, heterostructures, transition metal doping, single-atom construction, etc.). In addition, the catalytic mechanisms of the Pt-group catalysts, as well as their impacts on the OER and HER processes, are presented. These studies have made significant efforts to minimize the number of precious metals applied, enhance precious metal utilization, and lower the costs of electrolytic water.

Many challenges remain in developing electrocatalysts; further research in this area is necessary. Efforts in the following areas should be made in the following years: to develop a new type of catalyst with a unique structure and excellent catalytic performance; investigate the model catalyst’s reaction mechanism using in situ characterization and DFT simulations to understand the structure-activity relationship and help design the required catalyst; and as computer science progresses, machine learning and extensive data analysis may be investigated for the design and optimization of catalysts. We want to construct catalysts with high activity, selectivity, and stability by developing synthesis procedures, characterization techniques, and theoretical calculation methodologies. The precious metals in OER and HER processes can act as a new activity center, provide more active sites, improve electrical conductivity, adjust the electronic structure, optimize the adsorption energy barrier of intermediates, etc. Although fundamental understanding of the two half-reactions of electrolytic water has advanced fast, most recent investigations into reaction intermediates and reaction progression have been accomplished using density functional theory (DFT) simulations. As a result, there is an urgent need to develop new characterization techniques (e.g., in situ spectroscopy) and theoretical models that can guide the development of novel and efficient precious metal-based catalysts. The design and production of single-atom catalysts can considerably enhance precious metal utilization efficiency and minimize costs. However, because single atoms have a high surface energy, the increase in metal loading may cause agglomeration. As a result, developing highly dispersible single-atom catalysts with high loading content will hasten the advancement of the PEM water electrolysis technique. The efficiency of both OER and HER for complete water electrolysis is restricted, and many electrocatalysts are active for just one-half reaction, which is not helpful to the practical development of water electrolysis devices. As a result, there is an urgent need to produce highly efficient, highly active, and stable bifunctional catalysts capable of simplifying water electrolysis devices and procedures. Pt-group metals offer more incredible promise for this purpose than most other materials, with solid activity for both OER and HER.

Non-precious metals and non-metal catalysts struggle to maintain high activity and long-term stability in PEM water electrolysis systems due to the highly acidic conditions. The only catalysts that can meet the demands for long-term stable operation under such challenging conditions are precious metal-based catalysts. Compared to the commercial Pt/C and IrO_2_ catalysts, precious metal-based materials reduce the number of precious metals used and their HER and OER also show good catalytic activity during the electrolysis of water. Related previous studies are critical for guiding the development of new experimental strategies and the construction of advanced electrocatalysts with exceptional water electrolysis performance. However, research on precious metal-based materials in electrolytic water is still less systematic. The development of innovative precious metal-based catalysts requires significant effort. Reducing precious metal loading, improving the utilization of precious metals, and prolonging catalytic lifetime are crucial directions for creating a new generation of efficient, long-lasting, and cost-effective acidic water electrolysis catalysts.

## Figures and Tables

**Figure 1 nanomaterials-12-02618-f001:**
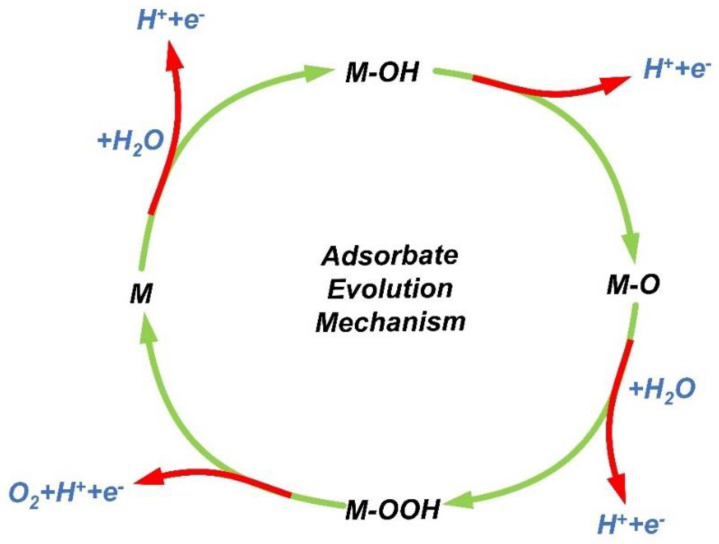
A typical AEM process in acid media is illustrated schematically.

**Figure 2 nanomaterials-12-02618-f002:**
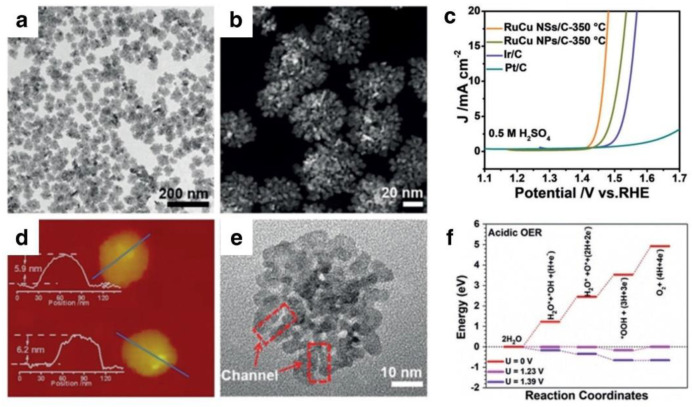
(**a**) TEM image, (**b**) HAADF–STEM image, (**c**) LSV curves of RuCu NSs/C−350 °C, RuCu NPs/C−350 °C, Ir/C, and Pt/C in 0.5 M H_2_SO_4_, (**d**) AFM images and corresponding thickness, (**e**) high–magnification TEM image, and (**f**) Reaction pathway of acidic OER on RuCu NSs. Reprinted with permission from Ref. [53], Copyright 2019, *John Wiley and Sons*.

**Figure 3 nanomaterials-12-02618-f003:**
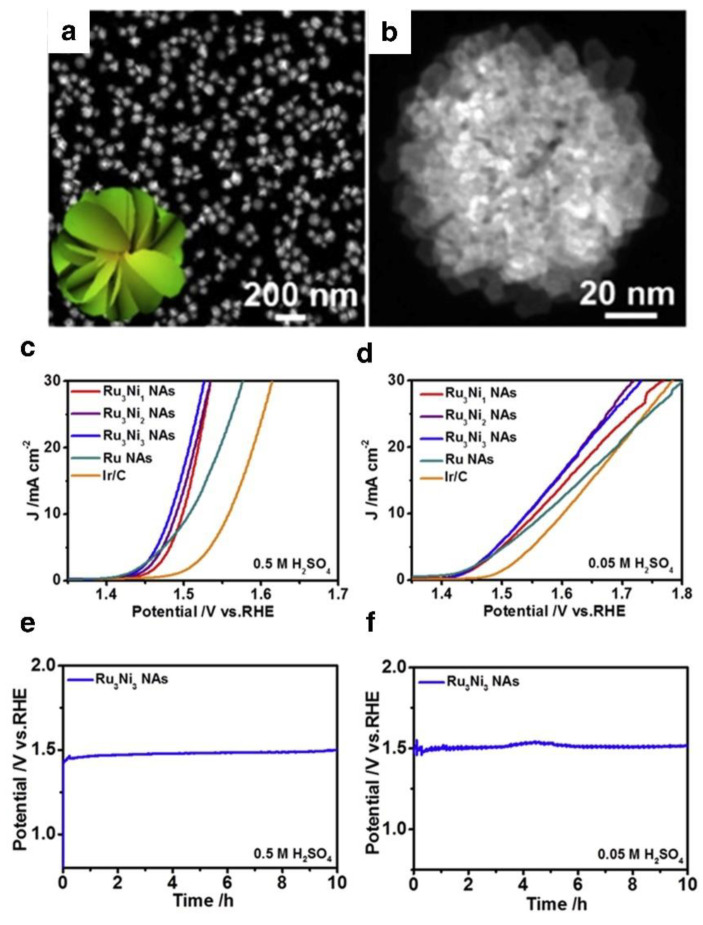
(**a**,**b**) HAADF–STEM image of the Ru_3_Ni_3_ NAs. LSV curves of the Ru_3_Ni_3_ NAs, Ru_3_Ni_2_ NAs, Ru_3_Ni_1_ NAs, Ru NAs, and Ir/C under different acidic conditions, (**c**) in 0.5 M H_2_SO_4_, (**d**) in 0.05 M H_2_SO_4_. Life test of the Ru_3_Ni_3_ NAs in (**e**) 0.5 M H_2_SO_4_ and (**f**) 0.05 M H_2_SO_4_ solutions at 5 mA cm^−2^. Reprinted with permission from Ref. [62], Copyright 2019, *Cell Press*.

**Figure 4 nanomaterials-12-02618-f004:**
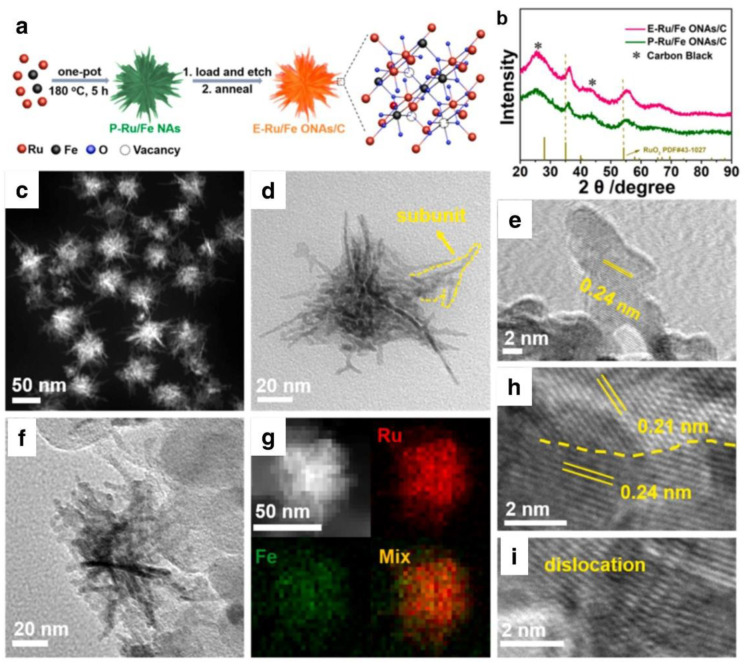
(**a**) Schematic depiction of the production of E–Ru/Fe ONAs, (**b**) XRD pattern, and (**c**) high–magnification TEM picture of P–Ru/Fe NAs, (**d**) High–magnification TEM picture, (**e**) HAADF–STEM–EDS elemental mappings, (**f**) HAADF–STEM image and (**g**–**i**) HRTEM images of E-Ru/Fe ONAs. Reprinted with permission from Ref. [61], Copyright 2021, *Elsevier*.

**Figure 8 nanomaterials-12-02618-f008:**
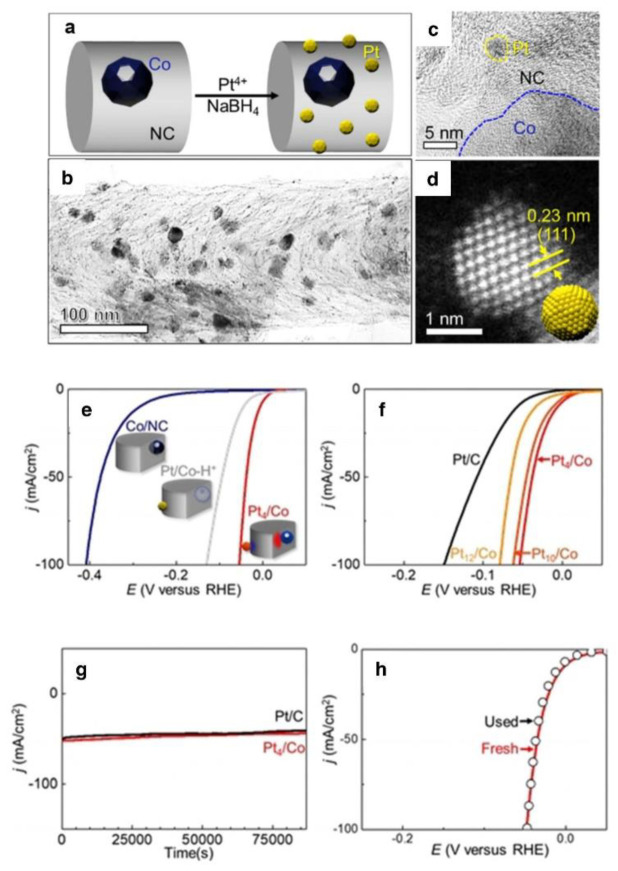
(**a**) Scheme demonstrating the synthesis process for depositing Pt nanoparticles on Co/NC nano-heterojunction materials, (**b**) TEM image, (**c**) HRTEM image, (**d**) HAADF--STEM image of a typical Pt_4_/Co sample, (**e**) LSV curves of the catalysts, (**f**) LSV curves for the Pt_x_/Co electrodes, (**g**) life tests of Pt_4_/Co and Pt/C electrodes, and (**h**) LSV curves of the Pt_4_/Co electrode before and after use for 24 h. All electrodes with the same catalyst loadings of 1 mg cm^2^ on carbon cloth (1 × 1 cm) were measured in Ar-saturated 0.5 M H_2_SO_4_ at a scan rate of 10 mV s^−1^. Reprinted with permission from Ref. [103], Copyright 2021, *John Wiley and Sons*.

**Figure 12 nanomaterials-12-02618-f012:**
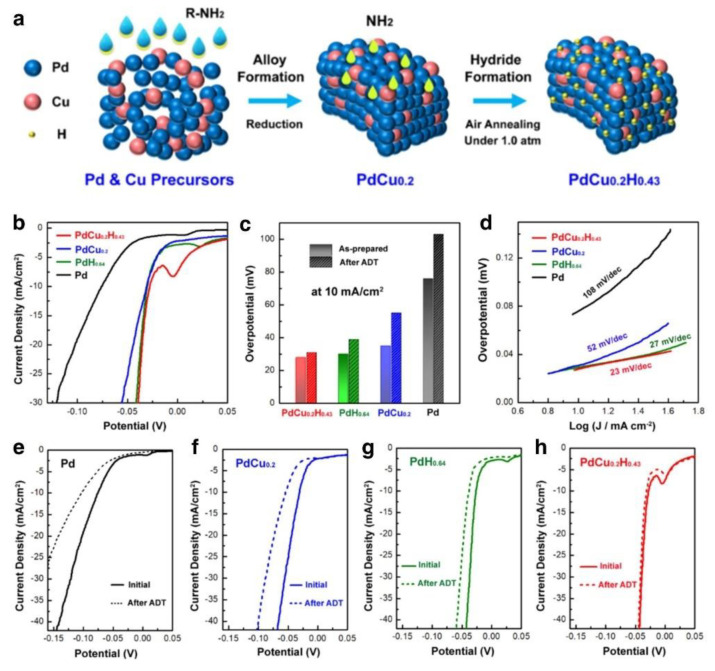
(**a**) Scheme of the PdCu_0.2_H_0.43_ nanoparticle formation, (**b**) LSV curves, (**c**) overpotentials at 10 mA/cm^2^, and (**d**) Tafel plots of these catalysts in 0.5 M H_2_SO_4_. LSV curves before and after 5000 cycles in 0.5 M H_2_SO_4_ for (**e**) Pd/C, (**f**) PdCu_0.2_/C, (**g**) PdH_0.64_/C, and (**h**) PdCu_0.2_H_0.43_/C. Reprinted with permission from Ref. [95], Copyright 2022, *American Chemical Society*.

**Figure 13 nanomaterials-12-02618-f013:**
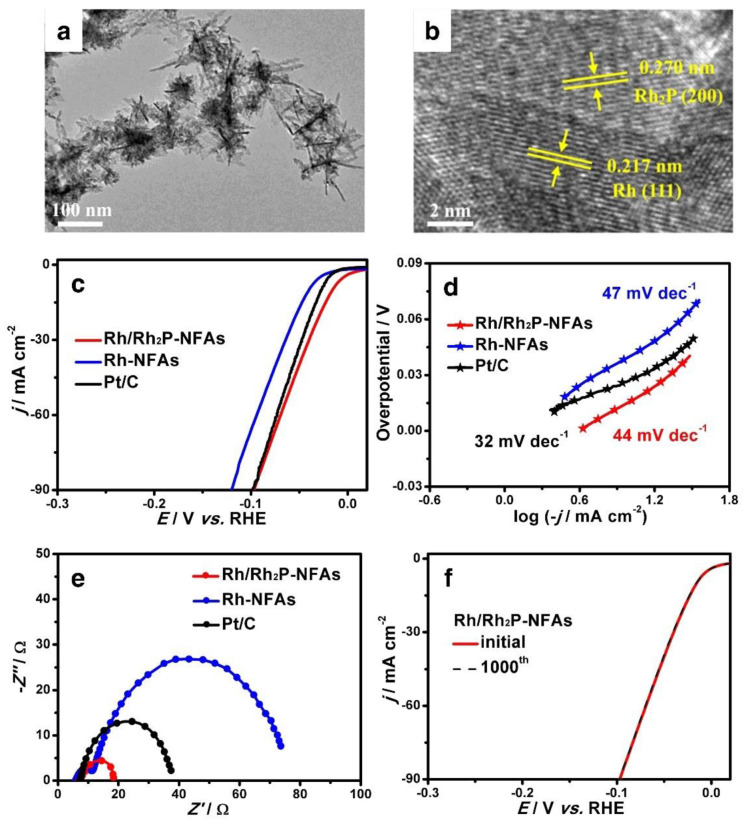
(**a**) TEM image of the Rh/Rh_2_P–NFAs, (**b**) HR-TEM image of the Rh/Rh_2_P–NFAs, (**c**) HER polarization curves at 5 mV s^−1^ without iR–correction in 0.5 M H_2_SO4, (**d**) Tafel slopes and (**e**) Nyquist plots at their open circuit potential. (**f**) The HER polarization curves of the Rh/Rh_2_P–NFAs before and after 1000 cycles at 5 mV s^−1^. Reprinted with permission from Ref. [135], Copyright 2021, *Elsevier*.

**Table 2 nanomaterials-12-02618-t002:** Summary of the HER performance for the electrocatalysts in an acidic medium.

Catalyst	Electrolyte	Activity at 10 mA/cm^2^ (mV)	Stability	Refs.
Ir@N-G-600	0.5 M H_2_SO_4_	27.8	1000 cycles	[49]
Ir_3_CeOx/C	0.5 M H_2_SO_4_	23	100 h@10 mA/cm^2^	[52]
RuCu NSs/C-350 °C	0.5 M H_2_SO_4_	19	20.5 h@5 mA/cm^2^	[53]
Ru-RuO_2_@NPC	0.5 M H_2_SO_4_	2	2000 cycles	[56]
Ir_6_Ag_9_ NTs	0.5 M H_2_SO_4_	2	1000 cycles	[60]
Ru@C_2_N	0.5 M H_2_SO_4_	13	1000 cycles	[86]
Pt-MoO_3_-x NFs | MoS_2_	0.5 M H_2_SO_4_	69	100 h@10 mA/cm^2^	[87]
Pd-CoCNTs	0.5 M H_2_SO_4_	2	3000 cycles	[88]
RuIr@NrC	0.5 M H_2_SO_4_	9	2000 cycles	[89]
Ru@Co-NC	0.5 M H_2_SO_4_	23	3000 cycles	[90]
Rh uNSs	0.5 M H_2_SO_4_	10	40 h@10 mA/cm^2^	[91]
IrP_2_-rGO	0.5 M H_2_SO_4_	8	20 h@10 mA/cm^2^	[92]
Co-Ir/C	0.5 M H_2_SO_4_	44	30 h@50 mA/cm^2^	[93]
MoO_x_-Pd/CC	0.5 M H_2_SO_4_	25	168 h@10 mA/cm^2^	[94]
PdCu_0.2_H_0.43_	0.5 M H_2_SO_4_	28	5000 cycles	[95]
RheRh_3_Se_4_/C	0.5 M H_2_SO_4_	32	22 h@50 mA/cm^2^	[96]
Pt–MoO_2_/MWCNTs	0.5 M H_2_SO_4_	68	2000 cycles	[97]
Au@Rh@PEI MNs	0.5 M H_2_SO_4_	30	24 h@10 mA/cm^2^	[98]
PtRh DNAs	0.5 M H_2_SO_4_	27	2000 cycles	[99]
a-Ru@Co-DHC	0.5 M H_2_SO_4_	27.8	28 h@−0.028 V_RHE_	[49]
α-Ni(OH)_2_@Ir	0.5 M H_2_SO_4_	20	50 h@10 mA/cm^2^	[100]
Ir NPs/siloxene	0.5 M H_2_SO_4_	31	1000 cycles	[101]
Ru-NiFeP/NF	0.5 M H_2_SO_4_	29	–	[102]
Pt_x_/C	0.5 M H_2_SO_4_	6.9	24 h@50 mA/cm^2^	[103]
Pt_3_Fe/BN	0.5 M H_2_SO_4_	38	3000 cycles	[104]
EA Pd–Co@Pd NPs–NF	0.5 M H_2_SO_4_	57	80 h@10 mA/cm^2^	[105]
Au@mRh NWs	0.5 M H_2_SO_4_	30	2000 cycles	[106]
Pd_45_@Ir_55_	0.1 M HClO_4_	11	1000 cycles	[107]
Cu/Rh(SAs) + Cu_2_Rh(NPs)/GN	0.5 M H_2_SO_4_	8	500 h@10 mA/cm^2^	[108]

## Data Availability

Not applicable.

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
