# Peer review of "Recent Advances Regarding Precious Metal-Based Electrocatalysts for Acidic Water Splitting"

_nanomaterials, 2022, doi:10.3390/nano12152618_

Round 1

Reviewer 1 Report

This reviewer paper overviewed recent advances on precious metal-based electrocatalysts for acidic water splitting. The topic is very imminently useful for audiences in the related fields and the references reviewed by the authors seem adequate. The manuscript is well organized with proper contents too. In this regard, I recommend this material be accepted if the authors clearly address the following questions in their revised manuscript.

Comment and questions:
1. There are numerous typos and grammatical errors, which must be corrected and revised the warrant the publication of the manuscript.
2. To capture a broader readership, the authors need to add suggestions on what issues should be further studied to boost this research area in the last section.

Reviewer 2 Report

The review submitted by Tian Tian, Haolin Tang et al analyzes some of the recent advances on the development of catalysts for acidic electrochemical water splitting. The manuscript is reasonably well organized, reviewing some of the precious metal-based electrocatalysts reported in recent years. I find the selected examples of reviewed compounds adequate. In fact, the authors could include other additional catalysts, since this is a field with a vast development and interest, in which multiple research lines are advancing, which supposes an added difficulty in choosing the examples to comment on.

The authors could pay more attention in their discussion to the development of single-atom catalysts (SACs). In the manuscript, there is only a reference to the Ir SACs. From my point of view, SACs represent an innovative strategy to minimize the usage of precious metals without compromising the catalytic performance. This strategy allows the reduction of metal loads since SACs contain isolated PGMs atoms singly dispersed on a catalyst support. On the other hand, catalysts based on bi-functional metal components and robust electrode substrates could also be discussed in some way.

In any case, I would accept the organization of the review and the choice of models described. What should certainly be corrected in the manuscript is the writing of some paragraphs, which are really difficult to understand, as well as a thorough revision of the language by an English native speaker.

Reviewer 3 Report

This manuscript deals with reviewing the progress made in precious metal-based electrocatalysis, mostly in the last 5 years (all references are from this period, except for the seminal paper from Norskov et al. from 2015). The authors give a good introduction into both OER and HER, although the discussion on OER mechanisms on pages 4-6 is not very well written, even confusing at times (see specific comments below) and should definitely be rewritten by a native speaker. Otherwise, the review is quite logical and well presented, although some additions must be made to make the picture clearer (see below):

1)      The size of the text is constantly changing throughout the manuscript (for example, on page 2, the first 1/3 of the page is sized 11.5, while the rest is 10). A uniform style should be used.

2)       Electrochemical water splitting in and of itself is not a method for storing energy (as is written on page 2), rather for converting chemical energy from hydrogen-hydrogen bonds to electrical energy. This should be clarified for the reader.

3)      The comparison of proton conductivity to hydroxide anion conductivity as an analogue to PEM/alkaline electrolysers is confusing. The proton conductivity of proton exchange membranes is much lower than that of acidic water-based solutions.

4)      The paragraph on page 5 is very confusing (especially the authors’ interpretation of the Sabatier principle, which is incorrect in its current form) and should be rewritten by a native speaker.

5)      On page 8, the authors have written that lowering the hydrogen adsorption energy barrier will increase the OER activity of Ru-based catalysts. However, as they earlier write, adsorbed hydrogen does not take part in the OER. What is the reason then for increased activity? A recent paper by Larsen et al. might offer some insights (Ha, M.-A., & Larsen, R. E. (2021). Multiple Reaction Pathways for the Oxygen Evolution Reaction May Contribute to IrO2 (110)’s High Activity. Journal of The Electrochemical Society, 168(2), 024506. https://doi.org/10.1149/1945-7111/ABDEEA)

6)      Page 14 is empty.

7)      Many of the best catalysts for the OER and HER incorporate carbon nanomaterials as support materials which can carry with them an additional CO2 penalty sometimes even comparable to the active material (especially in the case of MOFs), but they can also increase the activity of the active sites. Some discussion on the carbon effect and how novel carbon supports with minimal CO2 footprint can be used as supports should be added along with some topical references such as Oh, H. S., Nong, H. N., Reier, T., Bergmann, A., Gliech, M., Ferreira De Araújo, J., Willinger, E., Schlögl, R., Teschner, D., & Strasser, P. (2016). Journal of the American Chemical Society, 138(38), 12552–12563; Reier, T., Oezaslan, M., & Strasser, P. (2012). ACS Catalysis, 2(8) ; Remmel, A., Ratso, S., Divitini, G., Danilson, M., Mikli, V., Uibu, M., Aruväli, J., & Kruusenberg, I. (2021). ACS Sustainable Chemistry & Engineering, 10(1), 134–145;  Liu, Y., Chen, N., Li, W., Sun, M., Wu, T., Huang, B., Yong, X., Zhang, Q., Gu, L., Song, H., Bauer, R., Tse, J. S., Zang, S.-Q., Yang, B., & Lu, S. (2022 SmartMat, 3(2), 249–259; Murthy, A. P., Madhavan, J., & Murugan, K. (2018). Journal of Power Sources, 398, 9–26; Ratso, S., Walke, P. R., Mikli, V., Ločs, J., Šmits, K., Vītola, V., Šutka, A., & Kruusenberg, I. (2021). Green Chemistry, 23(12), 4435–4445.

Reviewer 4 Report

I strongly encourage the authors to discuss more extensively the paragraph on the application of MOF in the HER and OER.
